# OpenReview forum: "SORRY-Bench: Systematically Evaluating Large Language Model Safety Refusal Behaviors"
_NeurIPS.cc/2024/Datasets_and_Benchmarks_Track — Submitted to NeurIPS 2024 Track Datasets and Benchmarks_

### Official Review · Reviewer_KBhd · 2024-07-14
**Review for SORRY-Bench: Systematically Evaluating Large Language Model Safety Refusal Behaviors**

**Rating:** 7
**Confidence:** 3
**Correctness:** The paper appears to be correct in it…

**Review:**

This paper proposed SORRY-Bench, which is a benchmark designed to evaluate aligned large language models (LLMs) on their ability to recognize and reject unsafe user requests. By using a fine-grained taxonomy of 45 potentially unsafe topics and 450 class-balanced unsafe instructions compiled through human-in-the-loop methods, it can address the over-representation of certain topics seen in existing benchmarks. Besides, SORRY-Bench incorporates 20 diverse linguistic augmentations to systematically examine the effects of different languages, dialects, and prompt formats, ensuring a comprehensive evaluation. Additionally, the paper investigates design choices for creating a fast and accurate automated safety evaluator, demonstrating that fine-tuned 7B LLMs can achieve accuracy comparable to larger LLMs like GPT-4, but with lower computational costs. This effort evaluates over 40 proprietary and open-source LLMs on SORRY-Bench, analyzing their distinctive refusal behaviors and providing a systematic, balanced, and efficient approach to evaluating LLM safety refusal capabilities.

Pros:
The paper addresses a critical issue within its field of study.
The writing is clear, and the motivation is effectively conveyed in the introduction.
The experiment section is extensive and comprehensive, not only involving a significant workload but also well-designed to demonstrate the value of SORRY-Bench in the investigated research questions.
The discussion is engaging and thought-provoking.


Cons:
The details of the human annotation process in Section 2.2 need further elaboration. For instance, information about annotator agreement on manually labeling data to the 45-class safety taxonomy is missing in both the main text and the appendix.
Regarding the translation mutation methods, while SORRY-Bench explores several language options, it does not evaluate combinations of languages. It would be interesting to see how LLM performance shifts when dealing with inputs that include multiple languages.

**Strengths:**

The paper addresses a critical issue in its field of study and is well-written, clearly conveying its motivation in the introduction. The extensive and comprehensive experiment section demonstrates the value of SORRY-Bench, with a well-designed approach to investigating the research questions. Additionally, the discussion is engaging.

**Additional Feedback:**

N/A

**Clarity:**

The paper is clear and well-written, effectively conveying its motivation and findings.

**Documentation:**

The dataset is explicitly documented, and the code base is well-written.

**Ethics:**

There are no significant ethical concerns.

**Limitations:**

One limitation is the lack of detailed information on the human annotation process and annotator agreement. Additionally, the evaluation of language combinations in the translation mutation methods is not addressed.

**Opportunities For Improvement:**

There is an opportunity to elaborate more on the details of the human annotation process, particularly regarding annotator agreement on labeling data to the 45-class safety taxonomy. Additionally, evaluating the performance of LLMs with inputs containing multiple languages could provide valuable insights.

**Relation To Prior Work:**

Yes.

**Summary And Contributions:**

This paper proposed SORRY-Bench, which is a benchmark designed to evaluate aligned large language models (LLMs) on their ability to recognize and reject unsafe user requests. By using a fine-grained taxonomy of 45 potentially unsafe topics and 450 class-balanced unsafe instructions compiled through human-in-the-loop methods, it can address the over-representation of certain topics seen in existing benchmarks. Besides, SORRY-Bench incorporates 20 diverse linguistic augmentations to systematically examine the effects of different languages, dialects, and prompt formats, ensuring a comprehensive evaluation. Additionally, the paper investigates design choices for creating a fast and accurate automated safety evaluator, demonstrating that fine-tuned 7B LLMs can achieve accuracy comparable to larger LLMs like GPT-4, but with lower computational costs. This effort evaluates over 40 proprietary and open-source LLMs on SORRY-Bench, analyzing their distinctive refusal behaviors and providing a systematic, balanced, and efficient approach to evaluating LLM safety refusal capabilities.

---

> ### Author Rebuttal · Authors · 2024-08-15
>
> We would like to express our gratitude for the reviewer's efforts and time spent on composing this detailed and in-depth review, as well as the overall positive rating. We also apologize for any missing specifications or ambiguity in our paper. We hope our responses below could help address the reviewer's remaining questions and concerns.
>
> ---
>
> ### **1) The details of the human annotation process in Section 2.2 need further elaboration.**
>
> We apologize for any unclarity in our paper. Section 2.2 mainly documents our **methodology to curate the 45-class safety taxonomy**. There is actually no "human annotation" process during the taxonomy curation. As mentioned in Line 124-143, our taxonomy curation method consists of two stages:
> - In the first stage, we aggregate existing taxonomies from prior datasets. Then, we break down any vague and broad safety categories into more fine-grained categories, which makes our initial taxonomy.
> - In the second stage, we keep refining this intial taxonomy.
>     - First, we automatically map data points from these prior datasets to this initial taxonomy, using GPT-4 as a classifier. During the classification process, we set an "Others" category to capture data points that don't belong to any categories of the taxonomy.
>     - Then, we **manually review** all the data points classified as "Others" to decide if / how the initial taxonomy needs to be modified. Specifically, 4 authors manually go thorugh each of these data points, and decide how to update the taxonomy (e.g., add a new category, subdivide an existing category, and so on) after discussions.
>
>     The second stage is repeated multiple times, until all the 4 authors agree there is no need to further update the taxonomy.
>
> ---
>
> ### **2) Inter-annotator agreement regarding our human judgment dataset creation.**
>
> Meanwhile, the creation of our human judgment dataset (Sec 3.2) indeed involves human annotations. We have provided more details about the human annotation process in Appendix H.
>
> We have further validated the inter-annotator agreements among our 6 annotators, and show the results below. In specific, the 6 annotators are asked to label the same set of 100 model responses, which are randomly sampled from the 7.2K model responses from our human judge dataset, as either "Refusal" or "Compliance." Table below demonstrates the Cohen Kappa agreement between each pair of annotators. As shown, the pair-wise agreement is consistently above 86% (and >90% for most pairs), with an overall average agreement achieving **90.5%** (translating to 96.0% accuracy, 93.4% F1 score), indicating an almost perfect agreement among all annotators.
>
> | | **Annotator #1** |  **Annotator #2** | **Annotator #3** | **Annotator #4** | **Annotator #5** | **Annotator #6** |
> | --- | --- | --- | --- | --- | --- | --- |
> | **Annotator #1** | 1.00 | 0.88 | 0.91 | 0.90 | 0.90 | 0.93 |
> | **Annotator #2** | 0.88 | 1.00 | 0.88 | 0.97 | 0.92 | 0.86 |
> | **Annotator #3** | 0.91 | 0.88 | 1.00 | 0.90 | 0.90 | 0.89 |
> | **Annotator #4** | 0.90 | 0.97 | 0.90 | 1.00 | 0.95 | 0.88 |
> | **Annotator #5** | 0.90 | 0.92 | 0.90 | 0.95 | 1.00 | 0.88 |
> | **Annotator #6** | 0.93 | 0.86 | 0.89 | 0.88 | 0.88 | 1.00 |
>
> In future maintenance, we aim at further improving the quality of our human judge dataset via multiple annotations.
>
> ---
>
> ### **3) The impact of combining multiple languages.**
>
> In our current dataset, we have considered 20 linguistic mutations as augmentations to our base set, including translating English unsafe instructions into 5 other languages. Just as the reviewer, we believe the use of different languages (other than just English) is an important linguistic characteristic dimension in the real world.
>
> We also agree with the reviewer, that evaluating LLM safety performance when the input prompts contain multiple languages could be an interesting research problem to explore. However, unlike statically translating an English-version prompt into a single alternative language, transforming it into a version that contains a combination of multiple languages is more open-ended -- the combination can be done in many different ways. As a consequence, *it's probably not suitable to consider the mixture use of languages in a static dataset.*
>
> Moreover, modeling the mixture use of multiple languages could be an independent and intriguing research problem. For example, we may first need to understand what are the best ways to combine multiple languages in an unsafe prompt, or what are the common multi-lingual prompts that actual users would write in the wild. However, studying this is out of scope of our submission at this moment. Still, we realize this is an interesting and insightful problem to explore, and will include this into the future maintenance plan of our benchmark.
>
> As a reference, we refer reviewer and interesting readers to prior work like [1], where the authors have considered using a mixture of languages (English jailbreaking template + translated unsafe prompt) to intentionally jailbreak LLMs.
>
> [1] Multilingual Jailbreak Challenges in Large Language Models, ICLR 2024
>
> ---
>
> Again, we sincerely appreciate the reviewer's reviewing effort. Don't hesitate to let us know if you need any additional clarifications or information.

---

> > ### Comment · Reviewer_KBhd · 2024-08-30
> >
> > Thank you for your response. I don't have any other concerns.

---

### Official Review · Reviewer_2i7Q · 2024-07-24
**Good paper-decent contribution in LLM text-to-text safety evaluations.**

**Rating:** 6
**Confidence:** 4
**Clarity:** Yes

**Review:**

- Clarity: The paper is well-written, and quite clear throughout. In some places, it would be good to know more details such as in section 2.3, the data collection seems to need more explanation, eg, how did you come up with the novel class balancing samples?

- Originality: Even though the dataset is constructed on top of existing benchmarks, the work is fairly original as it improves the quality such as balanced set and human annotations.

Overall, I would say the paper would have a moderate impact on text-to-text model safety evaluations.

**Strengths:**

Here are the strengths:

- The paper is well-written.

- Creating a class-balanced dataset is always difficult, from my experience, for some categories it is hard to come up with samples while for others it is not at all easy.

- Evaluating the evaluators is an interesting step before using evaluators

- The LLM as judge (Table 1) and LLM as content generator (Figure 4 and Table 2) evaluations are quite extensive.

**Additional Feedback:**

NA

**Correctness:**

The construction of the dataset seems fine, I would not say the collected samples are very exciting e.g. uncovering loopholes in LLMs in edge cases.

**Documentation:**

Decent but not very satisfactory, please refer to the section opportunities For Improvement point 2.

**Limitations:**

Yes

**Opportunities For Improvement:**

- I looked at the datasets, seem okay. I believe a safety benchmark should contain both safe and unsafe prompts to evaluate an LLM more extensively. It would be much more useful if SORRY-Bench had included a balanced set of safe yet harmful-looking prompts. This is why many recent datasets such as XSTest and follow-up works have started focusing on both sides of the coin. An LLM that always refuses and keeps generating "I'm sorry" irrespective of whatever prompt it sees will perform nicely on SORRY-Bench, the primary reason being the unavailability of safe prompts in the evaluation set.

- It would be good to know more about how the dataset samples were created, who were the annotators, how much was the inter-annotator agreement, and see how this benchmark (besides imbalance) contrasts with scores provided by the benchmark datasets mentioned in Appendix Table 3.

**Relation To Prior Work:**

Yes

**Summary And Contributions:**

The primary contribution of this work is a new dataset named SORRYBench that has the following distinguishing factors from the existing datasets:

- A fine-grained taxonomy of 45 possible hazards as opposed to existing, more coarse-grained benchmarks.
- Each category has an equal number of samples in SORRYBench compared to other benchmarks widely skewed towards certain categories.

As an additional contribution, the paper also provides a comparison between different LLMs acting as a judge in different settings with fine-tuned versions on the curated dataset consistently outperforming CoT and Few-Shot versions.

---

> ### Author Rebuttal · Authors · 2024-08-15
>
> We appreciate the reviewer's efforts spent on reviewing service, as well as the overall acknowledgement of our work. While we agree with the reviewer's points on potential opportunities that may further improve our work, please let us provide additional clarifications and addtional details as requested.
>
> ---
>
> ### **1) Should contain both safe and unsafe prompts to quantify over-refusal.**
>
> > An LLM that always refuses and keeps generating "I'm sorry" irrespective of whatever prompt it sees will perform nicely on SORRY-Bench.
>
> Indeed, a LLM that always refuses any requests (e.g., https://www.goody2.ai) would be considered the "safest" when we take into account of all 45 categories. We agree that evaluating only on SORRY-Bench does not reflect how "nice" a model is. Instead, we would suggest putting together SORRY-Bench and language model capability / utility benchmarks (e.g., Chatbot Arena[1], MMLU[2]). While the "LLM that always refuses any requests" may demonstrate strong safety on SORRY-Bench (it's indeed quite safe!), it will undoubtedly score low on these capability benchmarks (and thus less preferred).
>
> In our paper, all the evaluated LLMs are the popular ones with good capability and utility, and definitely would not always keep generating "I'm sorry." For future models, developers can (and should) run evaluations on both safety benchmarks (SORRY-Bench) and utility benchmarks -- which is actually what LLMs developers are doing nowadays. For example, in Gemma-2 paper[3], Table 13 reports utility benchmark results, while Table 18 reports safety benchmark results.
>
>
> > It would be much more useful if SORRY-Bench had included a balanced set of safe yet harmful-looking prompts (over-refusal evaluation).
>
> We completely agree that model over-refusal is a crucial dimension to evaluate. We did not discuss this in our current paper as *it's afterall a quite different research problem*. To explore what's the best practice, it possibly takes another independent paper (like OR-Bench[4] and XSTest[5]) to discuss what should / should not be captured in an "over-refusal" benchmark, the best method to curate a taxonomy over-refusal, dataset collection, and so on. All these research questions are somehow orthogonal to the ones discussed in our work.
>
> For now, we would recommend users to evaluate model over-refusal behaviors on **OR-Bench**[4] and **XSTest**[5]. While our current focus predominantly lies in evaluating LLM's refusal behaviors on potentially unsafe instructions, we leave over-refusal evaluation as an intriguing feature to be implemented in future maintainence. We note that there are definitely opportunities to craft a systematic over-refusal benchmarks stemming from our insights curating SORRY-Bench (reuse the 45-class taxonomy, mutate the current data points into a "safe" version, etc.). But establishing such an over-refusal dataset will take equivalently more efforts and additional reflections.
>
>
>
> [1] Gemma 2: Improving Open Language Models at a Practical Size, arXiv 2024
>
> [2] Chatbot arena: An open platform for evaluating llms by human preference, ICLR 2024
>
> [3] Measuring massive multitask language understanding, ICLR 2021
>
> [4] OR-Bench: An Over-Refusal Benchmark for Large Language Models, arXiv 2024
>
> [5] XSTest: A Test Suite for Identifying Exaggerated Safety Behaviours in Large Language Models, ACL 2024
>
> ---
>
> ### **2) More specifications.**
>
> > How are the dataset samples created?
>
> As detailed in Appendix F.1, our base dataset was collected by 9 graduate students and scholars from our internal research group. Each collector is asked to collect 10 data points for each of their assigned 45 / 9 = 5 safety categories. Here, we provide further details on our dataset collection procedure:
> - All 9 collectors first went through a 30min training session (regarding the 3 goals mentioned in line 623-640).
> - Then, they were provided with the GPT-4 classified data points (3,656 in total) from the 10 prior datasets, and were encouraged to use these data points as a data bank.
> - In particular, we asked collectors to select qualified data points (line 623-640) from the data bank (and make necessary modifications) if possible.
> - When the data points were insufficient, collectors are asked to manually create additional novel unsafe instructions for that category. In this case, we encouraged the collectors to either compose new data points themselves, or utilize web resources (e.g., search engine or AI assistance).
>
> > Who are the annotators for the human judge dataset?
>
> As described in Appendix H, six authors were serving as annotators.
>
> > Inter-annotator agreement
>
> We have validated the inter-annotator agreements among our 6 annotators, and the results are shown below. In specific, the 6 annotators are asked to label the same set of 100 model responses, which are randomly sampled from the 7.2K model responses from our human judge dataset, as either "Refusal" or "Compliance." Table below demonstrates the Cohen Kappa agreement between each pair of annotators. As shown, the pair-wise agreement is consistently above 86% (and >90% for most pairs), with an overall average agreement achieving **90.5%** (translating to 96.0% accuracy, 93.4% F1 score), indicating an almost perfect agreement among all annotators.
>
> | | **Annotator #1** |  **Annotator #2** | **Annotator #3** | **Annotator #4** | **Annotator #5** | **Annotator #6** |
> | --- | --- | --- | --- | --- | --- | --- |
> | **Annotator #1** | 1.00 | 0.88 | 0.91 | 0.90 | 0.90 | 0.93 |
> | **Annotator #2** | 0.88 | 1.00 | 0.88 | 0.97 | 0.92 | 0.86 |
> | **Annotator #3** | 0.91 | 0.88 | 1.00 | 0.90 | 0.90 | 0.89 |
> | **Annotator #4** | 0.90 | 0.97 | 0.90 | 1.00 | 0.95 | 0.88 |
> | **Annotator #5** | 0.90 | 0.92 | 0.90 | 0.95 | 1.00 | 0.88 |
> | **Annotator #6** | 0.93 | 0.86 | 0.89 | 0.88 | 0.88 | 1.00 |
>
> In future maintenance, we aim at further improving the quality of our human judge dataset via multiple annotations.
>
> ---
>
> (to be continued)

---

> ### Author Rebuttal · Authors · 2024-08-15
>
> (continuing the response above)
>
> ### **3) How is this benchmark contrasted with scores provided by the benchmark datasets mentioned in Appendix Table 3?**
>
> We also agree with the reviewer that it'd be good to have a side-by-side visualization of LLM safety scores across different safety benchmarks. Below, we compare SORRY-Bench with 4 recent LLM safety benchmarks [1,2,3,4] over 24 LLMs. For SORRY-Bench, as we did in our major results (Fig 1), we report the *compliance rate* -- i.e., the percentage of unsafe instructions each LLM complies with. For other benchmarks, we directly cite their reported results for each LLM. Note that for SALAD-Bench, ALERT and StrongREJECT, we report the reverse, i.e., $100\%-score$, to reflect the unsafe rate (instead of safe rate reported in their original papers), for an easier interpretation. Overall, a **higher** number in the table reflects a **higher** compliance rate of the (potentially) unsafe instructions in each benchmark dataset.
>
> In general, these safety benchmarks manifest similar trends for different LLMs. For example, Claude and Llama-2 models are considered on the "safest" end, whereas Mistral and Zephyr-7b-beta models are considered more "unsafe." Worth of notice, scores of these LLMs on SORRY-Bench are ranging from 6.00% to 90.22%, demonstrating how SORRY-Bench can well distinguish the safety refusal performance of various LLMs, benefiting from a fine-grained and balanced design.
>
> | Model | SORRY-Bench | HarmBench [1] | SALAD-Bench [2] | ALERT [3] | StrongREJECT [4] |
> | -- | -- | -- | -- | -- | -- |
> | Claude-2.1 | 6.00% | 2.0% | \ | \ | \ |
> | Claude-2.0 | 7.78% | 2.0% | 0.23% | \ | \ |
> | Claude-instant-1.2 | 8.44% | 5.0% | \ | \ | \ |
> | GPT-3.5-turbo-1106 | 11.33% | 33.0% | 11.38% | 3.05% | \ |
> | Llama-2-70b-chat | 13.11% | 2.8% | 3.79% | 0.02% | 0% |
> | Llama-2-7b-chat | 14.89% | 0.8% | 3.49% | \ | \ |
> | Llama-2-13b-chat | 15.56% | 2.8% | 3.19% | \ | \ |
> | Gemma-2b-it | 20.22% | \ | 4.10% | \ | \ |
> | GPT-4-1106-preview | 25.78% | 9.3% | 6.51% | \ | \ |
> | GPT-4-0125-preview | 27.11% | \ | \ | 0.82% | \ |
> | GPT-3.5-turbo-0613 | 28.00% | 21.3% | \ | \ | 4% |
> | GPT-4-0613 | 29.78% | 21.0% | \ | \ | 3% |
> | Vicuna-13b-v1.5 | 33.11% | 19.8% | 54.10% | \ | \ |
> | Gemini-pro | 33.56% | 18.0% | 11.69% | \ | \ |
> | Vicuna-7b-v1.5 | 35.56% | 24.3% | 55.54% | 4.25% | \ |
> | Zephyr-7b-r2d2 | 36.00% | 14.2% | \ | \ | \ |
> | Qwen1.5-72b-chat | 36.67% | \ | 7.00% | \ | \ |
> | Chatglm3-6b | 37.11% | \ | 9.55% | \ | \ |
> | Yi-34b-chat | 41.56% | \ | 12.87% | \ | \ |
> | Mixtral-8x7B-Instruct-v0.1 | 56.44% | 47.3% | 23.85% | 1.78% | \ |
> | Mistral-7b-instruct-v0.2 | 67.33% | 46.3% | 19.86% | 24.55% | \ |
> | Dolphin-2.6-mixtral-8x7b | 85.33% | \ | \ | \ | 78% |
> | Zephyr-7b-beta | 87.33% | 65.8% | \ | 22.14% | \ |
> | Mistral-7b-instruct-v0.1 | 90.22% | \ | 54.13% | \ | \ |
>
> - **Note**: These benchmarks may have adopted different configurations (e.g., system prompt, sampling hyperparameters, definition of "safety" metrics, different datasets and taxonomies). Due to these differences, **we strongly caution against direct comparison across these benchmarks**.
>
> [1] HarmBench: A Standardized Evaluation Framework for Automated Red Teaming and Robust Refusal, ICML 2024
>
> [2] SALAD-Bench: A Hierarchical and Comprehensive Safety Benchmark for Large Language Models, ACL 2024
>
> [3] ALERT: A Comprehensive Benchmark for Assessing Large Language Models' Safety through Red Teaming, arXiv 2024
>
> [4] A STRONGREJECT for Empty Jailbreaks, ICLR 2024 R2-FM Workshop
>
>
> ---
>
> Again, we sincerely appreciate the reviewer's reviewing effort. Don't hesitate to let us know if you need any additional clarifications or information.

---

### Official Review · Reviewer_Duy6 · 2024-07-25
**A refined practice**

**Rating:** 6
**Confidence:** 3
**Correctness:** Correct.
**Clarity:** Yes

**Review:**

The paper presents a well articulated advancement to refusal benchmarking, but the value to the community in fixing the categorical imbalance in the benchmark is of less utility to safety engineering than would seem to be indicated by the imbalance problem. Old benchmarks could be statistically rebalanced, or given that this is a safety benchmark, the minimum of the assessed properties could be presented. Safety is generally a property of the "worst case," so taking the min could be better, or taking a weighting on the classes proportionate to the risks presented by those classes.

Pros

* A strong advancement in an important benchmark area
* More categories than older benchmarks in the space
* Strong ablative study of models
* Provides license for dataset. Another paper operating in a substantially similar space does not, which tips this one towards acceptance and that one towards rejection.

Cons

* This is a crowded space and it is not clear whether this is sufficiently different/better than other benchmarks in the space to warrant admission to a selective proceeding. In the absence of a fatal flaw or something I would suggest be used instead, I am inclined to overlooking the crowd and considering this the tallest entrant.
* Reliance on machine perturbation is likely to introduce some biases in coverage towards what LLMs are currently likely to output. Uncovering or disproving this would take another research paper. This is not that research paper.

**Strengths:**

A needed advancement in the state of practice for refusal benchmarking. A worthwhile advancement and service to the community even if it does not present something fundamentally new and different.

**Additional Feedback:**

None

**Documentation:**

Yes

**Ethics:**

No.

**Limitations:**

Yes.

**Opportunities For Improvement:**

There is a strong drive towards singular benchmarks, but works such as these should produce many benchmarks aligned to a variety of safety needs. The minimum value of a benchmark category for instance is often as important as the average.

**Relation To Prior Work:**

Yes

**Summary And Contributions:**

The authors present an expanded taxonomy of refusal categories and prepare a balanced dataset conforming to that taxonomy.

---

> ### Author Rebuttal · Authors · 2024-08-15
>
> We are grateful for the reviewer's thoughtful assessment of our work and the overall positive rating. Still, we would like to take the opportunity to address the potential concerns / limitations raised and provide further clarifications.
>
> ---
>
> > ... but the value to the community in fixing the categorical imbalance in the benchmark is of less utility ... Old benchmarks could be statistically rebalanced ...
>
> We note that our contribution in curating a balanced dataset is more than just "rebalancing" prior benchmark datasets. While we indeed started from prior datasets and selected qualified data points from them when possible, we found that for many categories, their data points are insufficient (less than 10 qualified prompts). Thus, to build our balanced benchmark dataset, we have to **create numerous novel unsafe instructions**. This cannot be done by just statistically rebalancing old benchmarks.
>
>
> > Safety is generally a property of the "worst case," so taking the min could be better, or taking a weighting on the classes proportionate to the risks presented by those classes ... The minimum value of a benchmark category for instance is often as important as the average.
>
> We totally agree with the reviewer that safety evaluations should take care of the "worst case." This is exactly one key motivation behind our benchmark design. We curate our safety taxonomy to be both fine-grained and extensive (on top of the union of 10 prior benchmark's taxonomies), since **we don't want to overlook any potential scenarios that models may cause safety risks**. Our dataset, built upon this taxonomy, should help capture and evaluate comprehensive safety risks across a wide spectrum. As we have shown in Figure 4 (benchmark results), **our benchmark allows convenient audit of model risks in each of the 45 safety categories** -- e.g., model developers can easily interpret in which (finegrained) safety categories the model is performing "the worst." And when model developers are curious about which specific unsafe instructions the model failed to refuse (or comply with), they can conduct further inspection in the instance-level results.
>
>
> We also agree that simply computing an overall average compliance rate across all 45 safety categories is not the best way to represent model safety performance. While the overall average compliance rate provides a coarse-grained quantification of model safety, it fails to highlight a model's worst-case performance in specific categories, particularly those considered high-risk by stakeholders. And this is exactly why, in Figure 4, we report *per-class* compliance rates alongside the overall compliance rate for each model. It's important to clarify that **Figure 4 is not intended as a "leaderboard,"** nor do we advocate for pursuing an absolutely low average compliance rate on SORRY-Bench. Instead, we strongly recommend that benchmark users conduct a nuanced, category-wise analysis, as demonstrated in Section 4.2, rather than focusing solely on the singular averaged score.
>
> ---
>
> Again, we sincerely appreciate the reviewer's reviewing effort. Don't hesitate to let us know if you need any additional clarifications or information.

---

### Official Review · Reviewer_iAGQ · 2024-07-28
**Good dataset and model on LLM refusals related to safety**

**Rating:** 6
**Confidence:** 5
**Clarity:** The paper is very clear and well writ…

**Review:**

Having a comprehensive dataset on unsafe prompts is very relevant for the LLM safety community. While there are many existing datasets, I appreciate the authors' proposal to have a fine-grained taxonomy on unsafe but also prompts that might break some community policies. Also collecting responses from different LLMs and manually annotating them for refusals is a very relevant task. This dataset can then be used to: 1) finetune LLMs for the task of detecting refusals / compliance or engagement in a prompt  ; 2) to provide a comprehensive study on how different LLMs engage with unsafe prompts in different categories.

At the same time, there are several missing pieces that I detail below and that I would have expected for in a dataset paper. For the results to be relevant, it is very important for the authors to respond to as many of them to be sure the results provided in the paper are valid and do not contain any biases.

**Strengths:**

* Very relevant problem and useful dataset for the LLM safety community
* I like the methodology used in the paper to develop the 45 categories taxonomy, but some aspects are unclear - for the human annotations and interventions were there any inter-rater / multiple raters to validate the results?
* The finetuned models to detect LLM refusals have very good results on the test set of SORRY-Bench and may be very useful for the LLM safety community as additional judges to LLM as a judge and using rules and heuristics
* The paper is generally well-written, easy to understand, and has dense information in the annexes. The results might be interesting for the LLM community at large.

**Additional Feedback:**

No additional feedback.

**Correctness:**

Using only 6 LLMs for the refusals dataset and 10 prompt per category can induce biases as already mentioned. This is the main problem that might influence the correctness of the results.

**Documentation:**

Yes, documentation is good.

**Ethics:**

The paper seems to follow all standard practices for an LLM safety papers.

**Limitations:**

Using only 10 prompts per category seems like an important limitations, I would have liked to see at least results on the different train-test splits mentioned in the weaknesses section.

But the limitations section addresses correctly most of the limitations of the current work.

**Opportunities For Improvement:**

* On my side, the main missing thing for the dataset to be reliable is the lack of multiple annotations and study of inter-rater agreement and also of disagreement if these cases exist - and I believe they do. And this is for extrablishing the methodology, choosing the 10 samples per class, labeling refusals from the 6 LLMs
* The dataset is rather small, containing only 10 examples per category therefore I think this might bring some bias - how were this 10 samples chosen? Are they really enough?
* Which are the 6 LLMs used to generate samples - maybe I missed this information, but I cannot find it in the paper.
* How is the train-test split done? I would like to see splits per LLM, per category and also per prompt in a category. Especially the per LLM is important, how do we know that results trained on 6 LLMs are also valid on other LLMs as refusals may be very different.
* I have seen no stats on the class (refusal vs compliance) distribution in general and per category (and maybe also per LLM)
* I would have liked to also see accuracy and F1 scores for refusal task

**Relation To Prior Work:**

Related work section is relevant.

**Summary And Contributions:**

The paper proposes a valuable dataset related to LLM refusals to engage in topics that are unsafe or may be considered not respecting human values, preferences, or policies established by the companies developing these LLMs. The dataset called SORRY-Bench contains not only prompts, by also responses from 6 different LLMs, on 45 different categories. Using this dataset, the authors study the compliance rate of a wide range of open and closed LLMs, and also the performance of using LLMs, including finetuning them on the train set of SORRY-Bench, to detect refusals.

While I think the problem is very relevant to the NLP safety community and the proposed dataset and models are very useful for safety research, there are a couple of things that need to be improved before publishing the paper. I would really like to improve my review score if the authors are able to solve at least some of the comments / weaknesses below in the revised version of the paper.

---

> ### Author Rebuttal · Authors · 2024-08-15
>
> We sincerely thank the reviewer for the overall positive rating and detailed review of our work. We also apologize for any missing details or potential ambiguity. Below, please let us provide more clarifications regarding your questions and concerns.
>
> ---
>
> ### **1) Lack of inter-rater agreement study & multiple annotations for the human judgment dataset**
>
> We have validated the inter-annotator agreements among our 6 annotators, and the results are shown below. In specific, the 6 annotators are asked to label the same set of 100 model responses, which are randomly sampled from the 7.2K model responses from our human judge dataset, as either "Refusal" or "Compliance." Table below demonstrates the Cohen Kappa agreement between each pair of annotators. As shown, the pair-wise agreement is consistently above 86% (and >90% for most pairs), with an overall average agreement achieving **90.5%** (translating to 96.0% accuracy, 93.4% F1 score), indicating an almost perfect agreement among all annotators.
>
> | | **Annotator #1** |  **Annotator #2** | **Annotator #3** | **Annotator #4** | **Annotator #5** | **Annotator #6** |
> | --- | --- | --- | --- | --- | --- | --- |
> | **Annotator #1** | 1.00 | 0.88 | 0.91 | 0.90 | 0.90 | 0.93 |
> | **Annotator #2** | 0.88 | 1.00 | 0.88 | 0.97 | 0.92 | 0.86 |
> | **Annotator #3** | 0.91 | 0.88 | 1.00 | 0.90 | 0.90 | 0.89 |
> | **Annotator #4** | 0.90 | 0.97 | 0.90 | 1.00 | 0.95 | 0.88 |
> | **Annotator #5** | 0.90 | 0.92 | 0.90 | 0.95 | 1.00 | 0.88 |
> | **Annotator #6** | 0.93 | 0.86 | 0.89 | 0.88 | 0.88 | 1.00 |
>
> Although the annotators are highly consistent with each other, we understand there might still be potential noises or unexpected biases caused by human mistakes and disagreements underlying the dataset. Multiple annotations are indeed a possible way to further improve the labeling quality and reduce such potential noises and biases. At the time of submission, we have not done this since:
> - The labeling is quite costly, even for our current scale. To obtain our current 7.2K annotations, each annotator went through and labeled 7.2K / 6 = **1.2K model responses**, spending approximately **6~10 hours** in total. Labeling such unsafe textual content could be difficult (need to read sentence-by-sentence to decide whether it's actually complying with the unsafe request) and brought significant mental burdens to all annotators. Such huge cost prevented us to scale up to multiple annotations at submission time.
> - On the other hand, as validated earlier, all annotators achieved almost perfect agreement (90%+ Cohen Kappa agreement) on the refusal judgment task, indicating that the noises during annotation could negligible.
>
> However, we totally agree that a possible improvement is to obtain multiple labels for each model responses. We will include this in our next maintenance plan for our SORRY-Bench human judgment dataset.
>
> ---
>
> ### **2) How were the 10 samples per class chosen in SORRY-Bench dataset? Are they enough?**
>
>
> As detailed in Appendix F.1, our base dataset was collected by 9 graduate students and scholars from our internal research group. Each collector is asked to collect 10 data points for each of their assigned 45 / 9 = 5 safety categories. Here, we provide further details on our dataset collection procedure:
> - All 9 collectors first went through a 30min training session (regarding the 3 goals mentioned in line 623-640).
> - Then, they were provided with the GPT-4 classified data points (3,656 in total) from the 10 prior datasets, and were encouraged to use these data points as a data bank.
> - In particular, we asked collectors to select qualified data points (line 623-640) from the data bank (and make necessary modifications) if possible.
> - However, we found for many (~40%) categories, **the data points from prior datasets were insufficient (fewer than 10 qualified samples)**. In these cases, collectors are asked to manually create additional novel unsafe instructions for that category. We encouraged the collectors to either compose new data points themselves, or utilize web resources (e.g., search engine or AI assistance).
>
>
> We understand that our inclusion of 10 samples per class might not capture the entire spectrum of in-the-wild unsafe prompts, and may inevitably bring potential bias. However, we argue that **this shall not bring additionally more bias compared to prior efforts**, as evidenced by the fact that prior datasets have even fewer than 10 samples in many of our safety categories (where they under-represent). And for those categories where these prior datasets over-represent, we have carefully selected 10 qualified data points that are diverse in terms of both themes and lengths. Additionally, we have generated **20 mutated versions of each prompt**, resulting in a total of 210 prompts per category (9.5K prompts in total) -- this further amplifies the prompt diversity and reduces bias underlying prompt formatting.
>
> Moreover, SORRY-Bench is intended as an iterative benchmark that can be **extended in the future maintenance** to scale up and include more samples. For the time being, the current size (450 samples for the base set) is **moderate and allows for efficient evaluation**, making it practical for iterative research and development.
>
> ---
>
> (to be continued)

---

> ### Author Rebuttal · Authors · 2024-08-15
>
> (continuing the response above)
>
> ### **3) For the human judgment dataset...**
>
> > Which are the 6 LLMs used to generate samples?
>
> As a matter of fact, to create our human judge dataset, we sampled diverse responses from 31 LLMs (not “6 LLMs”). The large number of models we sampled from help ensure the inherent diversity of our human judge dataset, and provide more promise that the trained judge model can be valid on other unseen LLMs.
>
> Yet, we understand the reviewer’s concern of “whether a judge model trained on some LLMs can generalize well on other unseen LLMs.” To quantitatively validate this, we conduct an **ablation study**, where we **partition all human annotations by LLMs**, as suggested by the reviewer. Specifically, we use 3,685 responses from 15 LLMs as the train set, and reserve the other 3,515 responses from the remaining 16 LLMs as the test set.
>
> Following the practice in our main paper (Sec 4.1), we fine-tune Mistral-7b-instruct-v0.2 on the train set. We find that even in this case (where the judge model was trained on only 15 LLMs), the judge model can still perform well on the 16 unseen LLMs we hold for test, achieving a **substantial agreement of 78.13%** (Accuracy=90.18%, F1 score=89.05%).
>
> This validates our presumption -- the judge model trained on our human judgment dataset should be likely valid for unseen LLMs. Meanwhile, we also understand the distribution of model responses (refusal or compliance) may shift over time, and this may hazard the validity of our refusal judgment setup. Therefore, we also plan to iterate and update our human judgment dataset with newer LLM responses (and annotations) in the future.
>
> > How is the train-test split done?
>
> As stated in Sec 3.2, for each of the 450 unsafe instruction, we collected human annotations for 16 different model responses (sampled from 31 LLMs). We then randomly split these 16 annotations into 10 + 6 annotations, for the train and test split, respectively. In total, this results in 10 * 450 = 4,500 records in the train split, and 6 * 450 = 2,700 records in the test split.
>
> The rationale behind this splitting is to ensure balance across different unsafe instructions – we expect the trained judge model to perform equally well for each of the unsafe instructions in our dataset.
>
> ---
>
> ### **4) Refusal v.s. compliance distribution (of human judgment dataset)**
>
> We aplogize for having omitted these stats in our paper. Among the human judgment dataset, 69.08% records are "Refusal" and 30.92% are "Compliance." A detailed breakdown of the distribution by category is shown below:
>
> | Category  | 1 | 2 | 3 | 4 | 5 | 6 | 7 | 8 | 9 | 10 | 11 | 12 | 13 | 14 | 15 | 16 | 17 | 18 | 19 | 20 | 21 | 22 | 23 |
> |--|--|--|--|--|--|--|--|--|--|--|--|--|--|--|--|--|--|--|--|--|--|--|--|
> | Refusal |76.88%|76.88%|77.50%|75.62%|75.62%|81.88%|70.62%|76.25%|75.62%|73.75%|72.50%|45.62%|73.12%|76.25%|65.00%|71.88%|73.12%|83.12%|71.88%|75.00%|81.25%|70.62%|78.75%
> | Compliance |23.12%|23.12%|22.50%|24.38%|24.38%|18.12%|29.38%|23.75%|24.38%|26.25%|27.50%|54.37%|26.88%|23.75%|35.00%|28.12%|26.88%|16.88%|28.12%|25.00%|18.75%|29.38%|21.25%
>
> | Category | 24 | 25 | 26 | 27 | 28 | 29 | 30 | 31 | 32 | 33 | 34 | 35 | 36 | 37 | 38 | 39 | 40 | 41 | 42 | 43 | 44 | 45 | Total |
> |--|--|--|--|--|--|--|--|--|--|--|--|--|--|--|--|--|--|--|--|--|--|--|--|
> | Refusal |70.62%|70.00%|68.12%|76.88%|69.38%|58.13%|51.88%|77.50%|57.50%|62.50%|48.12%|40.62%|73.75%|71.25%|75.62%|70.00%|69.38%|58.75%|59.38%|43.75%|68.75%|68.12%|69.08%|
> | Compliance |29.38%|30.00%|31.87%|23.12%|30.63%|41.88%|48.12%|22.50%|42.50%|37.50%|51.88%|59.38%|26.25%|28.75%|24.38%|30.00%|30.63%|41.25%|40.62%|56.25%|31.25%|31.87%|30.92%|
>
>
> ---
>
> ### **5) Accuracy and F1 scores (for Table 1)**
>
> Our meta-evaluation results with **accuracy** and **F1 scores** alongside Cohen Kappa agreement are shown below (corresponding to Table 1 and 5 in our paper).
>
> | Model (+Method)| Cohen Kappa (%) | Accuracy (%) | F1 Score (%) |
> |--|--|--|--|
> | GPT-4o | 79.4 | 91.7 | 89.7 |
> | GPT-4o (+CoT) | 75.5 | 90.4 | 58.5 |
> | GPT-4o (+Few-Shot) | 80.0 | 92.0 | 90.0 |
> | GPT-4o (+Fine-tuned) | \ | \ | \ |
> | GPT-3.5-turbo | 54.3 | 82.6 | 76.9 |
> | GPT-3.5-turbo (+CoT) | 39.7 | 78.0 | 46.0 |
> | GPT-3.5-turbo (+Few-Shot) | 61.3 | 84.0 | 80.7 |
> | GPT-3.5-turbo (+Fine-tuned) | **83.9** | 93.2 | 91.9 |
> | Llama-3-70b-instruct | 72.2 | 88.9 | 86.0 |
> | Llama-3-70b-instruct (+CoT) | 33.5 | 74.3 | 66.5 |
> | Llama-3-70b-instruct (+Few-Shot) | 74.9 | 89.8 | 87.4 |
> | Llama-3-70b-instruct (+Fine-tuned) | **82.8** | 92.8 | 91.4 |
> | Llama-3-8b-instruct | 40.6 | 73.8 | 70.2 |
> | Llama-3-8b-instruct (+CoT) | -50.7 | 16.4 | 16.1 |
> | Llama-3-8b-instruct (+Few-Shot) | 0.8 | 41.0 | 40.9 |
> | Llama-3-8b-instruct (+Fine-tuned) | **81.2** | 92.2 | 90.6 |
> | Mistral-7b-instruct-v0.2 | 54.8 | 83.4 | 76.9 |
> | Mistral-7b-instruct-v0.2 (+CoT) | 61.2 | 85.2 | 80.3 |
> | Mistral-7b-instruct-v0.2 (+Few-Shot) | 14.1 | 64.3 | 57.1 |
> | Mistral-7b-instruct-v0.2 (+Fine-tuned) | **81.3** | 91.8 | 90.6 |
> | Gemma-7b-it | 54.5 | 77.2 | 76.2 |
> | Gemma-7b-it (+CoT) | 43.5 | 78.5 | 71.4 |
> | Gemma-7b-it (+Few-Shot) | -54.6 | 17.4 | 16.2 |
> | Gemma-7b-it (+Fine-tuned) | **81.3** | 92.0 | 90.6 |
> |- | - | - | - |
> | Llama-3-70b (+Few-Shot) | 72.4 | 88.5 | 86.2 |
> | Llama-3-8b (+Few-Shot) | 22.8 | 63.7 | 60.7 |
> | Mistral-7b-v0.2 (+Few-Shot) | 71.6 | 88.3 | 85.8 |
> | Gemma-7b (+Few-Shot) | 64.3 | 83.1 | 81.8 |
> | Bert-Base-Cased (+Fine-tuned) | 75.0 | 89.1 | 87.5 |
> | Llama-Guard-2-8B | 39.7 | 75.8 | 69.8 |
> | MD-judge | 36.2 | 73.6 | 68.1 |
> | Perspective API | 1.0 | 70.1 | 42.5 |
> | Keyword Match | 38.1 | 72.1 | 68.8 |
>
> ---
>
> Again, we sincerely appreciate the reviewer's reviewing effort. Don't hesitate to let us know if you need any additional clarifications or information.

---

### Decision · Program_Chairs · 2024-09-26

**Decision:**

Reject

**Comment:**

This paper introduces a new LLM safety benchmark called SORRY-Bench, which evaluates LLM safety across 45 safety categories. The categories are aggregated and broken down from prior work, and the authors collect 10 unsafe prompts per category (from existing data + hand-written ones where missing), plus 20 mutations/augmentations of each prompt. The authors test a large number of LLMs on the benchmark, finding diverse safety issues.

Reviewer opinions on this paper are mildly positive, with scores of 6, 6, 6, and 7. Review quality is fairly high, with each review touching on different aspects of the paper. The authors provided extensive responses to all reviews. However, only one reviewer (KBhd) responded to the rebuttal, keeping their score.

I believe reviewer assessments are generally well-founded. In particular, I agree with KBhd’s concerns around a lack of detail on human data annotation, which plays a key role in both constructing the benchmark (assigning prompts to categories) and evaluating models on it (labelling responses as safe or not). Generally, I find that the exact nature and characteristics of the prompts across the 45 categories is hard to grasp from the paper. This is especially true given the abstract naming of some of the categories (“Governance Decision Advice” or “Evasion of Law”).

My biggest concern is around discrepancies in whether prompts actually are unsafe across safety categories. For example, #12: Impersonation, #34 and #35 are complied with by almost all models, including GPT-4, which makes me wonder if they truly are unsafe prompts. If some categories of prompts *are* safe to comply with while others are not, then this severely undermines the interpretability of results, since this would mean that sometimes a higher score is better and sometimes a higher score is worse. Relatedly, the authors find that Llama-3 is *more* compliant on the benchmark than Llama-2 (lines 310-312). This is inconsistent with most other work that finds Llama-3 to be *safer* than Llama-2 (as one would expect from a major model update). One potential remedy would be annotating all prompts for safety and dropping categories with prompts that are not clearly unsafe.

On the positive side, I would like to highlight the large number of experiments run for this paper, and the large amount of work done in general: the authors tested a *lot* of models, created many prompt variations, and ran a fairly large annotation task for validating their auto-evaluators.

Overall, however, I am leaning towards recommending rejection, because of the reviewer concerns outlined above as well as my own. For a D&B submission, I am not sufficiently convinced of the quality of the dataset. I do believe a future version of this paper would be a good fit for another AI conference or workshop. Please note that this recommendation is in line also with overall score distributions across papers in the D&B track.

 Minor notes:
- Reading the title of “Evaluating Safety Refusal Behaviours”, I was thinking the paper would be about *how* models refuse, like [here](https://arxiv.org/abs/2407.12043v1). I would consider just “Evaluating LLM Safety”, which is consistent with prior work.
- I would like to see in the Appendix a table with 1 example prompt for each of the 45 safety categories. Readers should be able to get a better qualitative sense of the dataset from the paper.
- “SorryBench” may be a more elegant name than “SORRY-Bench”.